# Association of Metabolically Healthy Obesity and Future Depression: Using National Health Insurance System Data in Korea from 2009–2017

**DOI:** 10.3390/ijerph18010063

**Published:** 2020-12-23

**Authors:** Yongseok Seo, Seungyeon Lee, Joung-Sook Ahn, Seongho Min, Min-Hyuk Kim, Jang-Young Kim, Dae Ryong Kang, Sangwon Hwang, Phor Vicheka, Jinhee Lee

**Affiliations:** 1Wonju College of Medicine, Yonsei University, Wonju 26426, Korea; s_ys_@naver.com (Y.S.); yeonnii96@gmail.com (S.L.); 2Department of Psychiatry, Wonju College of Medicine, Yonsei University, Wonju 26426, Korea; jsahn@yonsei.ac.kr (J.-S.A.); mchorock@yonsei.ac.kr (S.M.); mhkim09@yonsei.ac.kr (M.-H.K.); 3Department of Internal Medicine, Wonju College of Medicine, Yonsei University, Wonju 26426, Korea; kimjy@yonsei.ac.kr; 4Department of Precision Medicine, Wonju College of Medicine, Yonsei University, Wonju 26426, Korea; dr.kang@yonsei.ac.kr; 5Institute of AI and Big Data in Medicine, Wonju College of Medicine, Yonsei University, Wonju 26426, Korea; arsenal@yonsei.ac.kr (S.H.); phorvicheka@yahoo.com (P.V.)

**Keywords:** metabolically healthy obese phenotype, metabolic syndrome, obesity, depression

## Abstract

(1) Background: The health implications associated with the metabolically healthy obese (MHO) phenotype, in particular related to symptoms of depression, are still not clear. the purpose of this study is to check whether depression and metabolic status are relevant by classifying them into four groups in accordance with the MHO diagnostic standard. Other impressions seen were the differences between sexes and the effects of the MHO on the occurrence of depression. (2) Methods: A sample of 3,586,492 adult individuals from the National Health Insurance Database of Korea was classified into four categories by their metabolic status and body mass index: (1) metabolically healthy non-obese (MHN); (2) metabolically healthy obese (MHO); (3) metabolically unhealthy non-obese (MUN); and (4) metabolically unhealthy obese (MUO). Participants were followed for six to eight years for new incidences of depression. The statistical significance of the general characteristics of the four groups, as well as the mean differences in metabolic syndrome risk factors, was assessed with the use of a one-way analysis of variance (ANOVA). (3) Results: The MHN ratio in women was higher than in men (men 39.3%, women 55.2%). In both men and women, depression incidence was the highest among MUO participants (odds ratio (OR) = 1.01 in men; OR = 1.09 in women). It was concluded as well that, among the risk factors of metabolic syndrome, waist circumference was the most related to depression. Among the four groups, the MUO phenotype was the most related to depression. Furthermore, in women participants, MHO is also related to a higher risk of depressive symptoms. These findings indicate that MHO is not a totally benign condition in relation to depression in women. (4) Conclusion: Therefore, reducing metabolic syndrome and obesity patients in Korea will likely reduce the incidence of depression.

## 1. Introduction

Obesity is usually one of the metabolic syndrome conditions. It is a cluster of cardiometabolic abnormalities, including elevated high blood pressure, fasting blood glucose and dyslipidemia [1]. However, this does not mean that all obese persons are suffering from metabolic abnormalities, and there is some evidence that the impact of obesity on health can be kept away from individuals who comprise the metabolically healthy obese (MHO) phenotype [2,3]. It is interesting to note the health implications associated with this phenotype, even though there are no consistent results across studies and the health outcomes have not been examined [4,5,6,7,8].

Obesity and depression are essential factors of disease burden, but the evidence that proves that these two conditions are associated with one another is still not vivid. Even though recent studies including meta-analysis of prospective cohorts have proposed that having a greater body mass index (BMI) can increase the risk of depressive symptoms, several individual studies show that there is no relationship between obesity and depressive symptoms, while another group of individual studies show that greater BMI can reduce the risk of future mental health problems and of suicide [9,10,11,12]. Metabolic syndrome has a relationship with depression, independent of obesity [13]. The analysis of the relationship between depressive symptoms and the MHO phenotype sheds light on the association of obesity and depression.

Only a few studies have investigated the relationship between depressive symptoms and the MHO phenotype [14,15,16]. Two of them have shown that there is not an increased risk of depressive symptoms in MHO individuals followed for two years and ten years in comparison with healthy and non-obese individuals [14,16]. However, in another study, which was a pooled analysis of eight cross-sectional studies, it was shown that there is a moderately increased risk of depressive symptoms in obese individuals with advantageous metabolic profiles in comparison with healthy and non-obese individuals [15].

The purpose of this study is to check whether depression and metabolic status are relevant by dividing them into four groups in accordance with the MHO diagnostic standard. The differences between sexes and the effects of the MHO on the occurrence of depression were observed.

## 2. Materials and Methods

### 2.1. Study Population

In this retrospective study, we used a database given by the National Health Insurance Services-Health Screening (NHIS-HEALS) Cohort in Korea. The insurance system was set by the Korean government and covers about 97.2% of the residents. Individuals aged ≥40 years can have a general health-screening program every 2 years. The screening has included standardized self-reporting questionnaires on routine laboratory tests of blood and urine, anthropometric and blood pressure measurements, medical history and lifestyle behaviors. The cohort profile of the NHIS-HEALS is presented elsewhere [17]. Furthermore, the NHIS gave a research-specific database from the NHIS-HEALS in accordance with the conditions set by the researchers. Our research-specific database included 2009–2011 data of participants aged 19–69 years who had at least two general health-screening programs in 2009–2011. We extracted a list of participants from the research-specific database and excluded those who were aged ≤40 years or ≥70 years in 2009 or who did not participate in a general health screening program in 2009 (n = 4,708,511). Thus, all the participants in the list have their own 2009 health screening records. Participants who had one or more missing values in the metabolic syndrome (MetS) components were excluded (n = 9448) because MetS scores were not available. To exclude participants with depression, participants who were receiving medications for depression or who had the following the 10th revision of the international classification of disease (ICD-10) codes (as a main diagnosis or a sub-diagnosis at baseline) were not included: F32.0 to F34.9 (n = 822,603). Medication status was determined by prescription records. Based on the individual’s smoking information entered in the survey response, participants whose smoking information had changed or was missing were also excluded (n = 289,968). A total of 3,586,492 participants (1,936,582 men and 1,649,910 women) participated in this study (Appendix A). The institutional review board of Yonsei University, Wonju College of Medicine, Korea (IRB number: CR318350) approved a Waiver of Informed Consent for this study.

### 2.2. Measurements and Definitions

Healthcare institutions for screening were selected in accordance with the Framework Act on Health Examinations and the standard requirements for equipment, manpower and facilities. To lessen the measurement errors, the average values of all test data from laboratory between 2009 and 2011 were used. Values beyond the extreme outlier were considered as missing values. Height, weight and waist circumference were measured, and BMI was calculated with the formula BMI = kg/m^2^, where kg is a participant’s weight in kilograms and m^2^ is the square of the participant’s height in meters. Blood samples for serum glucose and total cholesterol (TC) level measurement were acquired following an overnight fast [18]. Low-density lipoprotein cholesterol (LDL-C) levels were calculated from TC, high-density lipoprotein cholesterol (HDL-C) and triglyceride (TG) levels or measured directly.

Obesity was defined as BMI ≥ 25 kg/m^2^ and metabolically healthy as metabolic syndrome risk < 2 in the 2009–2011 health-screening program. The participant’s level of alcohol consumption, frequency of physical activity, family economic status and smoking status were acquired using a set of questions. Smoking status was grouped as current smoker and not, and alcohol consumption as heavy drinker (i.e., consumption of 14 and 7 units of alcohol per week in men and women, respectively) and not. Regular exercise was interpreted as moderate to vigorous intensity physical activity for more than three days each week.

We categorized the participants into four main groups according to their metabolic status and BMI:Metabolically healthy non-obese (MHN): those who have less than two metabolic syndrome risk factors and a BMI under 25 kg/m^2^;Metabolically healthy obese (MHO): those who have less than two metabolic syndrome risk factors and a BMI of 25 kg/m^2^ or greater;Metabolically unhealthy non-obese (MUN): those who have more than two metabolic syndrome risk factors and a BMI under 25 kg/m^2^;Metabolically unhealthy obese (MUO): those who have more than two metabolic syndrome risk factors and a BMI of 25 kg/m^2^ or greater.

### 2.3. Study Outcome

In this study, we registered the population at risk between 2009 and 2011 and analyzed the outcomes in the follow-up period from 2014 to 2017, succeeding a 2-year washout period (2012–2013). The primary endpoint of the study was newly diagnosed depression in the follow-up period. Depression was determined by a recording of international classification of diseases (ICD)-10 codes F32.0 to F34.9 on health insurance data or the taking of an antidepressant (Appendix A). Medication status was determined by the Anatomical Therapeutic Chemical (ATC) code provided in the National Health Insurance Survey. 

### 2.4. Statistical Analysis

The statistical significance of the general characteristics of the four groups and the mean differences in metabolic syndrome risk factor were assessed with the use of one-way analysis of variance (ANOVA). The depression incidence among the four groups was assessed and compared with the odds ratio (OR) using multiple logistic regressions by complex sampling. We applied the multivariable-adjusted proportional hazards model: model 1 adjusted for age, while model 2 adjusted for age, alcohol consumption, exercise and smoking status. We also carried out a subgroup analysis based on the sex of the participants. We also compared the OR between seven metabolic syndrome risk factors adjusted for the participant’s age with the use of multiple logistic regressions.

## 3. Results

### 3.1. Baseline Characteristics of the Study Population

There were a total of 3,586,492 participants (1,936,582 men and 1,649,910 women) enrolled in this study. Table 1 and Table 2 show the baseline characteristics of both the men and women participants included in the analysis by BMI categories and metabolic status. The MHN ratio in women was higher than in men (men 39.3%, women 55.2%). According to the study, 11–12% of obese participants were described as metabolically healthy, i.e., with no more than one metabolic risk factor.

### 3.2. Relation between Metabolically Healthy Obesity and Depression

According to the pooled analysis for men participants with MHN as the reference category, a relationship with a higher risk of depressive symptoms was only shown in the MUO group (fully adjusted OR = 1.012; confidence interval (CI) = 1.002, 1.023) (Table 3).

In female participants, however, compared to MHN as the reference category, a higher risk of depressive symptoms presented in all three other groups (Table 4). The relationship with depressive symptoms was significantly higher for MUO (fully adjusted OR = 1.096; CI = 1.085, 1.107). In comparison to all non-obese participants (MHN or MUN), the depression risk for MUO (fully adjusted OR = 1.096; CI = 1.085, 1.107) was higher than for MHO (fully adjusted OR = 1.073; CI = 1.061, 1.086). Table 4 also indicates that, in comparison to all metabolically healthy participants (MHO or MHN), the depression risk for MUO (fully adjusted OR = 1.096; CI = 1.085, 1.107) was higher than for MUN (fully adjusted OR = 1.035; CI = 1.024, 1.046).

### 3.3. Relationship between Metabolic Syndrome Factors and Depression

Table 5 and Table 6 have shown the relationship between incident depression and metabolic syndrome factors for males (Table 5) and females (Table 6). In both sexes, the conclusion was that the greater the waist circumference, the greater the frequency of depression. In male participants, fasting blood sugar is also associated with depression (fully adjusted OR = 1.001; CI = 1.001, 1.001), while, in female participants, BMI is also associated with depression (fully adjusted OR = 0.994; CI = 0.994, 0.995).

## 4. Discussion

As far as we know, this is the first Korean population-based study to depict the relevance of both depression and metabolically healthy obesity. Even though recent studies indicate that metabolically healthy obesity (MHO) is related to an increased risk of depressive symptoms, there are still irregularities in the reports [14,15,16]. However, those analyses did not account for sex. In this study, we have found that the MHO group has a higher future depression risk than other subgroups in female participants, while, in male patients, there is a similar future depression risk to other subgroups.

The principal strength of this study is its nationally representative population-based study design with a huge pooled sample size. Our study is special and different from other results that can meta-analyzed based on the literature and biased by the selective publication of positive results because our current analysis was based on publicly available databases from National Health Insurance Database of Korea and not published results. It is logical to assume that the present results of these datasets generally represent Korea so they are not likely to be subject to a major publication bias.

Bi-directional associations have been outlined for the relationships between metabolic syndrome and depression, proposing that obesity, depressive symptoms and metabolic abnormalities could be associated through multiple pathways [9,19,20]. Using the NHIS-HEALS cohort, we have registered the population at risk between 2009 and 2011 and analyzed the outcomes in the follow-up period from 2014 to 2017. By excluding participants previously diagnosed with depression between 2009 and 2011, it is possible to analyze the temporal direction of the association.

In this study, complete case analysis was done by excluding participants who had one or more missing values in the MetS components (n = 9448) and whose smoking information had changed or was missing (n = 289,968). However, this study did not characterize the excluded population, which can result in bias.

The mechanisms that determine metabolically unhealthy and healthy obesity states are not popular [21,22]. One crucial factor could be where we should store the person’s fat, with excess visceral fat being more harmful for metabolic health than excess subcutaneous fat [3]. Additionally, some analysis has shown that people categorized as MHO have different health characteristics to those categorized as MUO, including higher physical activity, lower smoking prevalence and higher educational levels, proposing that both behavioral and physiological factors could be involved [15]. There are also various common biological states that link metabolic factors and obesity to depression, such as impaired glycemic control, inflammation and dysregulation of the hypothalamic–pituitary–adrenocortical axis [23,24,25,26,27,28]. A different set of factors may determine the depression risk of MHO individuals from non-obese individuals, such as negative self-image, physical inactivity, functional limitations in daily life, social stigma and discrimination [29,30,31].

The differences in future depression between metabolically healthy obesity men and women are still not known, but there are some studies on the different effects of sex of obesity and depression. One study proposed that prenatal stress-immune programming of the different sexes effects hypothalamic-pituitary-adrenal-gonadal axes and on metabolic and cardiac functions, leading to differences between the sexes in the comorbidity of major depressive disorders and obesity/metabolic syndrome [32]. Another study has shown that obesity has a relationship with different psychosocial profiles in both men and women [33,34]. Women are associated with being overweight and having an increased risk of suicidal tendencies and clinical depression, while men are the opposite [35]. Men may favor a large muscular body rather than a skinny one and having a high body weight may not increase the risk of depression as much as being underweight. Moreover, we also noted that the greater the waist circumference, the greater the frequency of depression. However, including BMI, the incidence of depression did not affect other metabolic syndrome factors.

## 5. Conclusions

In conclusion, the present results from a large pooled analysis of men and women show that MUO (metabolically unhealthy obesity) has a higher risk of depressive symptoms than MHN (metabolically healthy non-obese). Furthermore, in women participants, MHO (metabolically healthy obesity) is also related to a higher risk of depressive symptoms. These findings indicate that MHO is not a totally benign condition in relation to depression in women. Therefore, reducing metabolic syndrome and obesity patients in Korea will likely reduce the incidence of depression.

## Figures and Tables

**Table 1 ijerph-18-00063-t001:** Characteristics of the study population at the baseline (men).

	MHN	MUN	MHO	MUO
	(*n* = 760,561)	(*n* = 441,741)	(*n* = 213,940)	(*n* = 520,340)
Age (years)	50.3 ± 7.85	51.6 ± 7.78	49.4 ± 7.49	50.4 ± 7.64
Income classification				
Highest 25% (%)	13.5	16.2	11.8	15.0
Upper-middle 25% (%)	17.2	18.1	14.7	15.8
Lower-middle 25% (%)	25.9	26.4	24.2	25.4
Lowest 25% (%)	43.4	39.2	49.3	43.8
Alcohol consumption				
≥3/week (%)	20.3	28.9	19.3	26.2
2/week (%)	18.5	20.6	19.7	21.3
1/week (%)	26.2	22.4	26.8	23.4
<1/week (%)	35.1	28.2	34.3	29.1
Non-smokers (%)	24.1	19.5	25.9	21.2
Ex-smokers (%)	34.3	34.4	40.2	39.0
Current smokers (%)	41.6	46.1	33.9	39.8
Vigorous activity (%)	27.0	26.1	31.4	27.5
BMI (kg/m^2^)	22.3 ± 1.78	23.0 ± 1.51	26.3 ± 1.18	27.3 ± 1.89
Waist (cm)	79.6 ± 5.06	82.5 ± 4.88	86.7 ± 3.91	91.0 ± 5.34
FBS (mg/dL)	94.1 ± 14.57	110.6 ± 28.47	94.1 ± 12.44	108.9 ± 26.01
HDL (mg/dL)	56.3 ± 18.57	51.5 ± 24.10	53.1 ± 17.94	48.8 ± 19.97
TG (mg/dL)	112.8 ± 55.50	197.9 ± 111.05	127.1 ± 61.80	210.4 ± 117.76
sBP (mmHg)	112.0 ± 10.36	129.5 ± 11.46	122.3 ± 9.72	130.3 ± 11.19
dBP (mmHg)	75.1 ± 6.99	81.8 ± 7.30	76.5 ± 6.73	82.4 ± 7.55

Mean ± standard deviation or proportions of participants are indicated. Abbreviations: BMI, body mass index; dBP, diastolic blood pressure; FBS, fasting blood sugar; HDL, high-density lipoprotein; MHN, metabolically healthy non-obese; MHO, metabolically healthy obese; MUN, metabolically unhealthy non-obese; MUO, metabolically unhealthy obese; sBP, systolic blood pressure; TG, triglyceride. Vigorous activity is defined as physical activity more than three times a week with a strength of moderate, severe or above.

**Table 2 ijerph-18-00063-t002:** Characteristics of the study population at baseline (women).

	MHN	MUN	MHO	MUO
	(*n* = 910,641)	(*n* = 258,827)	(*n* = 198,719)	(*n* = 281,723)
Age (years)	49.5 ± 7.35	54.2 ± 7.90	51.4 ± 7.66	54.7 ± 7.94
Income classification				
Highest 25% (%)	27.9	26.7	28.8	26.7
Upper-middle 25% (%)	21.7	21.1	22.1	21.8
Lower-middle 25% (%)	20.7	22.9	22.1	24.0
Lowest 25% (%)	29.7	29.4	27.1	27.5
Alcohol consumption				
≥3/week (%)	3.1	3.1	3.3	3.1
2/week (%)	4.6	3.7	4.7	3.9
1/week (%)	15.4	11.0	14.7	11.4
<1/week (%)	76.9	82.2	77.4	81.6
Non-smokers (%)	97.0	96.6	97.5	96.8
Ex-smokers (%)	1.1	0.9	0.9	1.0
Current smokers (%)	1.9	2.4	1.6	2.2
Vigorous activity (%)	24.4	23.8	25.1	22.9
BMI (kg/m^2^)	22.0 ± 1.77	22.8 ± 1.54	26.6 ± 1.52	27.8 ± 2.35
Waist (cm)	73.0 ± 5.19	76.9 ± 5.42	81.6 ± 4.72	87.0 ± 6.08
FBS (mg/dL)	91.0 ± 10.98	104.8 ± 23.51	92.2 ± 10.78	105.3 ± 23.59
HDL (mg/dL)	62.4 ± 20.69	51.7 ± 21.93	60.6 ± 20.43	52.9 ± 20.98
TG (mg/dL)	90.4 ± 37.37	161.5 ± 81.53	101.1 ± 39.52	159.3 ± 79.25
sBP (mmHg)	116.1 ± 11.28	127.4 ± 12.84	120.3 ± 10.81	129.8 ± 12.37
dBP (mmHg)	72.2 ± 7.47	79.2 ± 8.08	74.3 ± 7.05	80.3 ± 7.86

Mean ± standard deviation or proportions of participants are indicated. Abbreviations: BMI, body mass index; dBP, diastolic blood pressure; FBS, fasting blood sugar; HDL, high-density lipoprotein; MHN, metabolically healthy non-obese; MHO, metabolically healthy obese; MUN, metabolically unhealthy non-obese; MUO, metabolically unhealthy obese; sBP, systolic blood pressure; TG, triglyceride. Vigorous activity is defined as physical activity more than three times a week with a strength of moderate, severe or above.

**Table 3 ijerph-18-00063-t003:** Odds ratio (OR) (95% CI) for the relationship between metabolic health and obesity with a risk of depression over three years of follow-up (men).

	Cases/N	Model 1 OR (95% CI)	Model 2 OR (95% CI)
Metabolically healthy non-obese (MHN)	104,143/760,561	1.000 (Ref)	1.000 (Ref)
Metabolically unhealthy non-obese (MUN)	64,297/441,741	1.012 (1.001–1.023)	1.009 (0.998–1.019)
Metabolically healthy obese (MHO)	28,149/213,940	0.999 (0.984–1.013)	1.002 (0.987–1.016)
Metabolically unhealthy obese (MUO)	72,235/520,340	1.014 (1.003–1.024)	1.012 (1.002–1.023)

Abbreviations: CI, confidence interval; Model 1: adjustment for age; Model 2: adjustment for age, alcohol consumption, exercise and smoking status.

**Table 4 ijerph-18-00063-t004:** OR (95% CI) for the relationship between metabolic health and obesity with a risk of depression over three years of follow-up (women).

	Cases/N	Model 1 OR (95% CI)	Model 2 OR (95% CI)
Metabolically healthy non-obese (MHN)	189,972/910,641	1.000 (Ref)	1.000 (Ref)
Metabolically unhealthy non-obese (MUN)	63,850/258,827	1.038 (1.027–1.049)	1.035 (1.024–1.046)
Metabolically healthy obese (MHO)	46,256/198,719	1.072 (1.060–1.085)	1.073 (1.061–1.086)
Metabolically unhealthy obese (MUO)	73,531/281,723	1.099 (1.088–1.110)	1.096 (1.085–1.107)

Abbreviations: CI, confidence interval; Model 1: adjustment for age; Model 2: adjustment for age, alcohol consumption, exercise and smoking status.

**Table 5 ijerph-18-00063-t005:** Relationship between metabolic syndrome factors and incident depression (men).

	Model 1	Model 2
	OR (95% CI)	OR (95% CI)
BMI	0.999 (0.998–1.001)	0.978 (0.976–0.981)
Waist	1.004 (1.003–1.004)	1.011 (1.010–1.012)
FBS	1.001 (1.001–1.001)	1.001 (1.001–1.001)
HDL	1.000 (1.000–1.000)	1.000 (1.000–1.001)
TG	1.000 (1.000–1.000)	1.000 (1.000–1.000)
sBP	0.997 (0.997–0.998)	0.997 (0.996–0.998)
dBP	0.996 (0.996–0.997)	0.999 (0.998–1.000)

Abbreviations: BMI, body mass index; CI, confidence interval; dBP, diastolic blood pressure; FBS, fasting blood sugar; HDL, high-density lipoprotein; OR, odds ratio; sBP, systolic blood pressure; TG, triglyceride. Model 1: adjustment for age; Model 2: adjustment for risk factors and age of metabolic syndrome.

**Table 6 ijerph-18-00063-t006:** Relationship between metabolic syndrome factors and incident depression (women).

	Model 1	Model 2
	OR (95% CI)	OR (95% CI)
BMI	1.013 (1.011–1.014)	0.995 (0.993–0.997)
Waist	1.008 (1.007–1.008)	1.010 (1.009–1.010)
FBS	1.001 (1.000–1.001)	1.000 (1.000–1.000)
HDL	1.000 (1.000–1.000)	1.000 (1.000–1.000)
TG	1.001 (1.001–1.001)	1.001 (1.000–1.001)
sBP	0.998 (0.998–0.998)	0.994 (0.994–0.995)
dBP	0.999 (0.999–1.000)	1.004 (1.004–1.005)

Abbreviations: BMI, body mass index; CI, confidence interval; dBP, diastolic blood pressure; FBS, fasting blood sugar; HDL, high-density lipoprotein; sBP, systolic blood pressure; TG, triglyceride. Model 1: adjustment for age; Model 2: adjustment for age and risk factors of metabolic syndrome.

## Data Availability

The cohort profile of the NHIS-HEALS is presented elsewhere [17].

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
