# Peer review of "Association of Metabolically Healthy Obesity and Future Depression: Using National Health Insurance System Data in Korea from 2009–2017"

_ijerph, 2020, doi:10.3390/ijerph18010063_

Round 1
Reviewer 1 Report
In this manuscript, ‘Association of Metabolically Healthy Obesity and Future Depression; Using National Health Insurance System Data in Korea from 2009-2017’, the authors studied the association between MHO and Depression occurrence risk. This is an exciting topic. The strengths of the study are
- Simple straight forward retrospective follow up study design
- A large sample size of around 3.5 million subjects
- Gender analysis and outcome.
- Statistical analysis with a multivariate-adjusted proportional hazards model adjusted for age, alcohol, exercise, and smoking.
- All the tables are informative and self-sufficient.
However, we cannot reach the conclusion the study indicates as there are some concerns as follows:
- Self-reported questionnaire.
- It is well established that obesity is related to depression in various studies published previously. The authors are dissecting obesity into 4 categories and tried to identify which MHO specifics are associated with depression risk overall. There are already published studies in this regard but at a smaller scale and lesser sample size.
- What about other factors such as economic status per family member, educational status, job status, marital status, family size-children, living with parents and in-laws, other disease conditions, and medicine consumption that may indirectly affect the depression. To understand the pure relationship between MHO and depression, there is a need to set a high level of exclusion criteria that might confound the statistical outcome. Please provide more details and reanalyze the data by excluding such indicators.
4. Rigorous English language editing is required. At many places it is not clear what the authors convey and there are lot of grammatical mistakes starting from the first sentence of the abstract. In abstract-conclusion- “ therefore, a multifaceted judgment and careful management were required to reduce metabolic syndrome and obesity and as well as to reduce the incidence of depression” does not make any sense at all. It is not aligned with the study design, methods or results at all.
Author Response
In this manuscript, ‘Association of Metabolically Healthy Obesity and Future Depression; Using National Health Insurance System Data in Korea from 2009-2017’, the authors studied the association between MHO and Depression occurrence risk. This is an exciting topic. The strengths of the study are
Simple straight forward retrospective follow up study design
A large sample size of around 3.5 million subjects
Gender analysis and outcome.
Statistical analysis with a multivariate-adjusted proportional hazards model adjusted for age, alcohol, exercise, and smoking.
All the tables are informative and self-sufficient.
However, we cannot reach the conclusion the study indicates as there are some concerns as follows:
Self-reported questionnaire.
(1) It is well established that obesity is related to depression in various studies published previously. The authors are dissecting obesity into 4 categories and tried to identify which MHO specifics are associated with depression risk overall. There are already published studies in this regard but at a smaller scale and lesser sample size.
What about other factors such as economic status per family member, educational status, job status, marital status, family size-children, living with parents and in-laws, other disease conditions, and medicine consumption that may indirectly affect the depression. To understand the pure relationship between MHO and depression, there is a need to set a high level of exclusion criteria that might confound the statistical outcome. Please provide more details and reanalyze the data by excluding such indicators.
Response: In this study, we used a database given by the National Health Insurance Services-Health Screening (NHIS-HEALS) Cohort in Korea. Although there were many other factors such as other disease conditions and medicine consumption, the data provided by the National Health Insurance Services-Health Screening (NHIS-HEALS) were entered as null values and could not be used for study.
Thank you for letting us know the shortcomings of our manuscript. To redeem this, we added the sentences and revised the manuscript as suggested in the revised manuscript. (Materials and Methods, lines 80-93):
“Our research-specific database included 2009–2011 data of subjects aged 19–69 years who had at least two general health-screening programs in 2009–2011. We extracted a list of subjects from the research-specific database and excluded those who were aged ≤40 years or ≥70 years in 2009 or who did not participate in a general health screening program in 2009 (n=4,708,511); thus, all the subjects in the list have their own 2009 health screening records. Subjects who had one or more missing values in the MetS components were excluded (n=9,448) because MS scores were not available. To exclude subjects with depression, subjects with receiving medications for depression or who had the following ICD-10 codes (as main diagnosis or sub-diagnosis at baseline) were not included: F32.0 to F34.9 (n=822,603). Medication status was determined by prescription records. Based on the individual’s smoking information entered in survey response, subjects whose smoking information has changed or missing were also excluded (n=289,968). A total of 3,586,492 subjects (1,936,582 men and 1,649,910 women) were participated in this study. (Supplement 1)”
(2) Rigorous English language editing is required. At many places it is not clear what the authors convey and there are lot of grammatical mistakes starting from the first sentence of the abstract. In abstract-conclusion- “therefore, a multifaceted judgment and careful management were required to reduce metabolic syndrome and obesity and as well as to reduce the incidence of depression” does not make any sense at all. It is not aligned with the study design, methods or results at all.
Response: Thank you for your opinion. Our description did not alined with the study and was rather confusing. As recommended, the paragraph in the Abstract-Conclusion section was revised as follows (Abstract, lines 39-40; Conclusion, lines 251-252):
“Therefore, a multifaceted judgment and careful management was required to reduce metabolic syndrome and obesity and as well as to reduce the incidence of depression. reducing metabolic syndrome and obesity patients in Korea will likely reduce the incidence of depression.”

Reviewer 2 Report
This is a very large study of the associations between a combination of risk factors for metabolic syndrome and depression in data from the Korean national routine health screening program.
Major comments
- The section on study outcome suggests that this is a cohort study in which risk factors for metabolic syndrome were determined in 2009-2011, and the occurrence of depression in 2014-2017. However, in the discussion, it is stated that the analysis was based on cross-sectional data, with the limitations on assessing direction of association pointed out. Therefore, the design of the study is completely unclear to me.
- To determine depression, a range of ICD codes is specified, and it is stated that an antidepressant had to be taken (I could not connect to the supplementary table to find out if this was through recording of reimbursement for prescribed medicines, or if thus was something entered in the coding of the health screenings). There is no description of how depression was determined in order for an ICD code to be entered. in the health screening, was this done by interviewer were participants asked to complete questionnaires. If questionnaires, did these vary? There is considerable variation in the performance of instruments to assess depression. See for example PMID: 25736983 and PMID: 30894161.
- Within the range of ICD codes, some refer to recurrent depression. Could splitting the codes for recurrent depression from the other codes help in considering direction of association. Also, it might be relevant to consider severity of depression
- The very large study size is a potential strength. However, I have no sense of what was done about missing data. Was the analysis a complete case analysis, which can be biased because missingness can be predictive of outcome. See PMID: 16980149
- Were the sets of questions to determine alcohol consumption etc (lines 97-99) validated?
- From the table for men, you can classify smoking as never, ex and current. Collapsing categories to create a binomial variable likely has resulted in loss of useful information.
Author Response
This is a very large study of the associations between a combination of risk factors for metabolic syndrome and depression in data from the Korean national routine health screening program.
Major comments
(1) The section on study outcome suggests that this is a cohort study in which risk factors for metabolic syndrome were determined in 2009-2011, and the occurrence of depression in 2014-2017. However, in the discussion, it is stated that the analysis was based on cross-sectional data, with the limitations on assessing direction of association pointed out. Therefore, the design of the study is completely unclear to me.
Response: Thank you for indicating the confusion of our manuscript. We used a database given by the National Health Insurance Services-Health Screening (NHIS-HEALS) Cohort in Korea. NHIS-HEALS is a cross-sectional survey, and we use two cross-sectional survey to confirm the association between obesity and depression. By excluding subjects previously diagnosed with depression, it is possible to analyze the temporal direction of the association. To clarify this, we removed the sentences and revised the manuscript. (Discussion, lines 216-222):
“The current analysis was done based on cross-sectional data, so it is not possible to analyze the temporal direction of the association. Longitudinal data shows that the relationship between depression and obesity is bi-directional, so that depression increases later obesity risk and obesity increases later depression risk [19]. Similar bi-directional associations have been outlined for relationships between diabetes and depression, and metabolic syndrome and depression, proposing that obesity, depressive symptoms, and metabolic abnormalities could be associated through multiple pathways [20,21]. Bi-directional associations have been outlined for the relationships between metabolic syndrome and depression, proposing that obesity, depressive symptoms, and metabolic abnormalities could be associated through multiple pathways [19-21]. Using NHIS-HEALS cohort, we have registered the population at risk between 2009 and 2011 and analyzed the outcomes in the follow-up period from 2014 to 2017. By excluding subjects previously diagnosed with depression between 2009 and 2011, it is possible to analyze the temporal direction of the association.”
(2) To determine depression, a range of ICD codes is specified, and it is stated that an antidepressant had to be taken (I could not connect to the supplementary table to find out if this was through recording of reimbursement for prescribed medicines, or if thus was something entered in the coding of the health screenings). There is no description of how depression was determined in order for an ICD code to be entered. in the health screening, was this done by interviewer were participants asked to complete questionnaires. If questionnaires, did these vary? There is considerable variation in the performance of instruments to assess depression. See for example PMID: 25736983 and PMID: 30894161.
Within the range of ICD codes, some refer to recurrent depression. Could splitting the codes for recurrent depression from the other codes help in considering direction of association. Also, it might be relevant to consider severity of depression
Response: We used a database given by the National Health Insurance Services-Health Screening (NHIS-HEALS) Cohort in Korea. Depression was determined by recording international classification of diseases (ICD)-10 code F32.0 to F34.9, or one has to take an antidepressant (supplement 2, 3 in this file). In the National Health Insurance database, the medical records table shows ICD-10 code and Anatomical Therapeutic Chemical (ATC) code. This questionnaire is not a questionnaire about symptoms of depression, but a questionnaire that confirms that depression was diagnosed in a hospital and does not include various questionnaires. To clarify this, we added the sentences and revised the manuscript. (Materials and Methods, lines 126-129):
“Depression was determined by recording international classification of diseases (ICD)-10 code F32.0 to F34.9, and or one has to take an antidepressant (supplement 2, 3). Medication status was determined by the Anatomical Therapeutic Chemical (ATC) code provided in the National Health Insurance Survey.”
(3) The very large study size is a potential strength. However, I have no sense of what was done about missing data. Was the analysis a complete case analysis, which can be biased because missingness can be predictive of outcome. See PMID: 16980149
Response: Thank you for indicating the limitations of our description. Subjects containing null values among the data were excluded from the study. Since this population was randomized and removed without specific criteria, it was not biased towards the results. To complement this, we added the sentences and revised the manuscript. (Materials and Methods, lines 85-87, 90-91):
“Subjects who had one or more missing values in the MetS components were excluded (n=9,448) because MS scores were not available.”
“Based on the individual’s smoking information entered in survey response, subjects whose smoking information has changed or missing were also excluded (n=289,968).”
(4) Were the sets of questions to determine alcohol consumption etc (lines 97-99) validated?
response: The subject’s amount of alcohol consumption, frequency of physical activity, family economic status, and smoking status were acquired using a set of questions, and this screening were set in accordance with the Framework Act on Health Examinations. In the case of alcohol consumption, the data provided indicates how many days of the week each individual was drinking alcohol. According to this, the subjects were divided into 4 groups; 3~7 days, 2 days, 1 day and less than 1 day per week.
(5) From the table for men, you can classify smoking as never, ex and current. Collapsing categories to create a binomial variable likely has resulted in loss of useful information.
response: Based on the individual’s smoking information entered in survey response, subjects whose smoking information has changed or missing were also excluded. If the smoking status of each individual was provided in pack-year from the data provided, it could be used as continuous variable by using the average value like other variables such as waist circumference or BMI. However, in the data provided, smoking status was only provided as a discrete variable of choosing one answer out of three, so it was best to divide it into three categories; Non-smokers, Ex-smokers, Current smokers.

Round 2
Reviewer 2 Report
Responses to previous comments appreciated.
The design seems to be based on the participants having taken part in the screening program in the period 2009-2011 and again in 2014-2017. This is still not fully explained in the methods, and I had to infer this from reading the discussion. The design should be more clearly explained in the methods and a flow diagram would be very helpful.
It would be very useful to have information on how the outcomes were distributed according to (1) specific ICD-10 codes , i.e., not just lumping together F32.0-F34.9; (2) specific code and medication; (3) medication only (and if so, what). For example, is MetS predominantly associated with MDD, mild depression etc?
It would also be helpful if the way in which depression was ascertained were further clarified in the methods and considered in the discussion. You state in your response to the first review that "a questionnaire that confirms that depression was diagnosed in a hospital" was used but in the manuscript you state that data came from a variety of standardized self-report questionnaires, including one or more relating to medical history and lifestyle behaviors within the general health-screening program. This is still very confusing. Resolving this communication is important as there is a great deal of criticism of previous studies of the ethology of depression.
From your response to my questions about missing information, I understand that a complete case analysis was done. This can result in bias, as is well explained in the reference I suggested in my previous comments (PMID: 16980149 https://pubmed.ncbi.nlm.nih.gov/16980149/ ). Almost 300,000 people were excluded because of missing information on smoking history. How did these people, who would otherwise have been included in the study, differ from the participants? Such a comparison would inform possible use of imputation, and interpretation of the results. Overall, over 9000 people were excluded because of missing information on MetS - again, how do their characteristics compare with participants?
As a general comment, the term "subjects" should be avoided - see https://www.ncbi.nlm.nih.gov/pmc/articles/PMC1115535/
The paper needs to be reviewed and revised by a native English speaker
Author Response
(1) The design seems to be based on the participants having taken part in the screening program in the period 2009-2011 and again in 2014-2017. This is still not fully explained in the methods, and I had to infer this from reading the discussion. The design should be more clearly explained in the methods and a flow diagram would be very helpful.
Response: In this study, participants were selected through screening program in 2009-2011, and depression diagnosed in those participants was confirmed in 2014-2017. You can check this through study outcomes. (Materials and Methods, lines 79-81, 123-125):
“Our research-specific database included 2009–2011 data of participants aged 19–69 years who had at least two general health-screening programs in 2009–2011.”
“In this study, we have registered the population at risk between 2009 and 2011 and analyzed the outcomes in the follow-up period from 2014 to 2017, succeeding a 2-year washout period (2012–2013). The primary endpoint of the study was a newly diagnosed depression in the follow-up period.”
(2) It would be very useful to have information on how the outcomes were distributed according to (1) specific ICD-10 codes , i.e., not just lumping together F32.0-F34.9; (2) specific code and medication; (3) medication only (and if so, what). For example, is MetS predominantly associated with MDD, mild depression etc?
Response: Thank you for your opinion with which we strongly agree. Unfortunately, we did not have information on how the outcomes were distributed according to specific ICD-10 codes and medication.
In study, patients with at least one prescription for antidepressants were also included in the diagnosis of depression, referring to previous studies. (https://doi.org/10.4093/dmj.2017.41.4.296)
(3) It would also be helpful if the way in which depression was ascertained were further clarified in the methods and considered in the discussion. You state in your response to the first review that "a questionnaire that confirms that depression was diagnosed in a hospital" was used but in the manuscript you state that data came from a variety of standardized self-report questionnaires, including one or more relating to medical history and lifestyle behaviors within the general health-screening program. This is still very confusing. Resolving this communication is important as there is a great deal of criticism of previous studies of the ethology of depression.
Response: Sorry to confuse you with the first review. As you know, we used a database given by the National Health Insurance Services-Health Screening (NHIS-HEALS) Cohort in Korea. Depression was determined not by a questionnaire, but by the hospital-diagnosed ICD code on the health insurance data. Meanwhile, the participant’s amount of alcohol consumption, frequency of physical activity, family economic status, and smoking status were acquired using a set of questions. To clarify this, we revised the manuscript as follows. (Materials and Methods, lines 105-107, 125-126):
“The participant’s amount of alcohol consumption, frequency of physical activity, family economic status, and smoking status were acquired using a set of questions.”
“Depression was determined by recording international classification of diseases (ICD)-10 code F320 to F349 on the health insurance data, or one has to take an antidepressant (Supplement 2, 3).”
(4) From your response to my questions about missing information, I understand that a complete case analysis was done. This can result in bias, as is well explained in the reference I suggested in my previous comments (PMID: 16980149 https://pubmed.ncbi.nlm.nih.gov/16980149/ ). Almost 300,000 people were excluded because of missing information on smoking history. How did these people, who would otherwise have been included in the study, differ from the participants? Such a comparison would inform possible use of imputation, and interpretation of the results. Overall, over 9000 people were excluded because of missing information on MetS - again, how do their characteristics compare with participants?
Response: We absolutely agree with your opinion that missing information can result in bias. Unfortunately, there is currently no data to characterize the excluded population. We will describe this as a limitation of this study. (Discussion, lines 218-221):
“In this study, complete case analysis was done by excluding participants who had one or more missing values in the MetS components(n=9,448) and whose smoking information has changed or missing(n=289,968). However, this study did not characterize the excluded population, this can result in bias.”
(5) As a general comment, the term "subjects" should be avoided - see https://www.ncbi.nlm.nih.gov/pmc/articles/PMC1115535/
Response: We absolutely agree with your opinion that the term "subjects" should be avoided. We thank you for your kindly letting us amend it. As recommended, we modified subjects to participants. (Materials and Methods, lines 79-92, 99-101, 106-108, 126-127; Results, lines 148, 156, 216-217):
“Our research-specific database included 2009–2011 data of subjects participants aged 19–69 years who had at least two general health-screening programs in 2009–2011. We extracted a list of subjects participants from the research-specific database and excluded those who were aged ≤40 years or ≥70 years in 2009 or who did not participate in a general health screening program in 2009 (n=4,708,511); thus, all the subjects participants in the list have their own 2009 health screening records. Subjects Participants who had one or more missing values in the MetS components were excluded (n=9,448) because MetS scores were not available. To exclude subjects participants with depression, subjects participants with receiving medications for depression or who had the following ICD-10 codes (as main diagnosis or sub-diagnosis at baseline) were not included: F32.0 to F34.9 (n=822,603). Medication status was determined by prescription records. Based on the individual’s smoking information entered in survey response, subjects participants whose smoking information has changed or missing were also excluded (n=289,968). A total of 3,586,492 subjects participants (1,936,582 men and 1,649,910 women) were participated in this study (Supplement 1).”
“Height, weight, and waist circumference were measured, and BMI was calculated with the formula BMI = kg/m2 where kg is a subject’s participant's weight in kilograms and m2 is the square of the subject’s participant’s height in meters.”
“And, we also compared the OR between seven metabolic syndrome risk factors adjusted for the subject’s participant’s age with the use of multiple logistic regressions.”
“There are a total of 3,586,492 subjects participants (1,936,582 men and 1,649,910 women) enrolled in this study.”
“Mean ± standard deviation or proportions of subjects participants are indicated”
“By excluding subjects participants previously diagnosed with depression between 2009 and 2011, it is possible to analyze the temporal direction of the association.”
“The subject’s participant’s amount of alcohol consumption, frequency of physical activity, family economic status, and smoking status were acquired using a set of questions.”
(6) The paper needs to be reviewed and revised by a native English speaker
Response: We thank you for your kindly letting us amend it. The manuscript has been carefully reviewed by an experienced editor whose first language is English and who specializes in editing papers written by scientists whose native language is not English. Following your recommendation, we reviewed the manuscript with careful attention with grammatical errors and revised it once again.
